# Analysis of White and Dark without Pressure in a Young Myopic Group Based on Ultra-Wide Swept-Source Optical Coherence Tomography Angiography

**DOI:** 10.3390/jcm11164830

**Published:** 2022-08-18

**Authors:** Huimin Yu, Huan Luo, Xian Zhang, Jinfu Sun, Zheng Zhong, Xufang Sun

**Affiliations:** Department of Ophthalmology, Tongji Hospital, Tongji Medical College, Huazhong University of Science and Technology, 1095 Jie-Fang Road, Wuhan 430030, China

**Keywords:** ultra-wide SS-OCTA, WWOP, DWOP, myopia, peripheral degeneration

## Abstract

Purpose: This study aimed to investigate the incidence of white without pressure (WWOP) and dark without pressure (DWOP) in a young myopic group based on multimode imaging and to explore the quantitative changes in DWOP based on ultra-wide swept-source optical coherence tomography angiography (SS-OCTA). Methods: A total of 138 patients with high myopia (SE < −6.00 D) were recruited. Examinations, including indirect ophthalmoscope, ultra-wide color fundus photograph, and ultra-wide SS-OCTA, were conducted for each eye. A total of 50 of the 138 patients were selected for further analysis since their DWOP lesions in SS-OCTA could be well quantified. Results: The incidence rates of WWOP and DWOP in the young myopic group were 35.24% and 29.96%, respectively. The patients with a lower spherical equivalent (SE) showed a tendency to have a higher axial length (AL) and higher prevalence of WWOP. Multivariate regression analysis illustrated that a more serious SE and a longer AL were risk factors for both WWOP and DWOP. Eyes with DWOP lesions had lower vessel density (VD) in the whole retina (*p* < 0.001) and a deep vascular complex (DVC) (*p* < 0.001), and lower thickness of the outer retina (*p* < 0.001) compared with healthy counterparts. Conclusion: Ultra-wide SS-OCTA provided new insights into myopic-related peripheral retinal degenerations. DWOP was characterized by thinning of the outer retina and lower perfusion in DVC.

## 1. Introduction

Myopia is becoming progressively more prevalent in east Asian countries. High myopia, defined as SE < −6 D, is correlated with various kinds of maculopathy and peripheral retina degeneration. The prevalence of myopia in young adults is around 80–90% in East Asia [1]. The increasing prevalence of high myopia in young adults may lead to a heavy burden on society and increasing risk of widespread problems with low vision and blindness.

Not only myopic maculopathy but also myopic-related peripheral degenerations (MRPDs) are of importance to consider in teenagers and young adults [2]. For example, lattice degeneration is relatively common and may require prophylactic laser retinopexy, which is associated with retinal atrophy and may lead to retinal detachment [3]. The early detection and appropriate treatment of MRPDs has important clinical significance. White without pressure (WWOP) is more common in the myopic population, with an incidence rate of 46.5–51.7% [4,5,6]. Dark without pressure (DWOP) is also reported in myopic eyes, and can be concurrent with WWOP [7]. Moreover, the pathological significance of WWOP and DWOP in the development of myopia remains poorly understood, and ongoing clinical observation is advocated [7] The multimodal imaging of WWOP and DWOP revealed completely opposite changes in the photoreceptor reflectivity, but questions concerning how they formed and whether they influenced the local microcirculation are still worth investigating.

Optical coherence tomography angiography (OCTA) is an automatic and accurate stratification method for the diagnosis of retinal and choroidal diseases. In the past few years, the quantitative function of OCTA has provided a powerful method to observe macular and peripapillary microcirculation changes in the early stages of high myopia [8,9]. However, previous studies have mainly focused on changes in a small area, usually 3 mm × 3 mm or 6 mm × 6 mm, centered macula or optic, which is insufficient for investigating MRPDs.

Very recently, ultra-wide swept-source OCTA (SS-OCTA) has attracted much attention and has been applied to detect peripheral retina lesions [10,11,12]. Its high-quality imaging and reproducible scanning make it possible to quantify the retina and choroid [13]. Thus, this study aimed to investigate the incidence of WWOP and DWOP in young adults, evaluate the reproducibility of the SS-OCTA measurements, and explore the quantitative changes of lesions based on ultra-wide SS-OCTA.

## 2. Methods

### 2.1. Participants

This cross-sectional study was approved by the medical ethics committee of Tongji Hospital, Tongji Medical College, Huazhong University of Science and Technology. The study adhered to the Declaration of Helsinki guidelines. All of the participants gave written informed consent. The study was registered at http://www.chictr.org.cn (registration number ChiCTR2100043611, accessed on 1 March 2021).

We recruited 138 patients from Tongji Hospital from March 2021 to December 2021. The inclusion criteria were: (1) age between 18 and 30 years old; (2) best-corrected visual acuity more than 0.8; (3) spherical equivalent (SE) less than –6.00 D; (4) normal intraocular pressure; (5) clear refractive medium; (6) no history of previous ocular surgery; (7) no ocular diseases other than high myopia.

All participants underwent detailed ocular examinations, including intraocular pressure assessment (IOP) (NT-510, NIDEK Co., Ltd., Tokyo, Japan), refractive error assessment (AR-310A, NIDEK, Tokyo, Japan), slit-lamp biomicroscopy (BP900, Haag-Streit International, Köniz, Switzerland), indirect ophthalmoscope (YZ6H, 66Vision.Tech, Suzhou. China), and axial length (AL) (AL-scan, NIDEK Co., Ltd., Tokyo, Japan). The spherical equivalent (SE) was calculated as the sphere value plus half the cylindrical power.

### 2.2. Assessment of Dark and White without Pressure

After the initial screening of the indirect ophthalmoscope, all participants underwent ultra-wide color fundus photography (CFP) (Daytona, Optos, Dunfermline, UK) and swept-source optical coherence tomography (SS-OCT) (VG200S; SVision Imaging, Luoyang, Henan, China). According to previous research, white without pressure (WWOP) was defined as a hyperpigmented lesion region in CFP and OCT imaging, and was characterized as a hyperreflective EZ (Figure 1A) [7,14]. Dark without pressure (DWOP) was defined as a hypopigmented lesion region in CFP and OCT en-face images, and the border of DWOP corresponded to the site where the ellipsoid zone (EZ) faded or disappeared (Figure 1B) [7,15].

### 2.3. SS-OCTA Image Acquisition and Processing

The SS-OCTA (VG200S; SVision Imaging, Henan, China) was equipped with an SS laser with a central wavelength of about 1050 nm and a scanning rate of 200,000 A-scans per second. The axial resolution, lateral resolution, and scan depth were 5 μm, 13 μm, and 3 mm, respectively. The volume data for the SS-OCTA were collected using a raster scan procedure that consisted of 1024 (horizontal) and 1024 (vertical) B-scans that were repeated four times and averaged. To remove eye motion artifacts, the system included an eye tracking utility based on an integrated confocal scanning laser ophthalmoscope.

In eyes with DWOP or WWOP, SS-OCTA was performed in five different regions using the 12 mm × 12 mm scanning mode (Figure 2). The same scanning operation was also performed on the other eye of the same individual. All images were scanned by the same trained examiner (H.L.). Then, we performed quality evaluation and control of the acquired images to exclude non-conforming ones.

The segmentation software automatically detected the boundaries of the retinal layers from the structural OCT cross-sectional images (Figure 3B) [13]. The inner retina was defined from 5 μm above the internal limiting membrane (ILM) to 25 μm below the inner nuclear layer (INL), which was combined with the superficial vascular complex (SVC) and DVC. The inner two-thirds and outer one-third interfaces of the ganglion cell layer and inner plexiform layer (GCL + IPL) were used to segment the SVC and DVC. The outer retina was defined from 25 μm below INL to 10 μm above Bruch’s membrane (BM). The choroid was defined from 10 μm above BM to the chorioscleral junction. Before being quantified, all automatically segmented images were checked by a trained examiner (H.L.), and manually modified if the segmentation was wrong.

Then, we adjusted the scan size of these images by Littmann’s method [16] and Bennett’s formula [17], which account for magnification discrepancies among the eyes caused by different ALs.

### 2.4. SS-OCTA Image Quantification of DWOP Lesion Areas

In our study, WWOP lesions were all beyond the OCTA scanning area, so we concentrated on exploring the changes in anatomical appearance and quantitative indicators of each layer in the DWOP lesions.

The DWOP lesion sample size was determined by n_p_ = [(Z_α_ + Z_β_) σ_d_/ES]^2^ [18] in the Power Analysis and Sample Size (PASS) software 2020, using preliminary data obtained in our laboratory with the following assumptions: an α (type I error *p* value significance level) of 0.05 (two-tailed), a power (1 − β) of 90%, difference (ES) in the thickness of all retina between the DWOP lesion area and healthy control area of 6 μm, and a standard deviation (σd) of 10 μm. Considering the situation of loss of follow-up, we set the loss ratio of follow-up to 20%. Therefore, a minimum of 37 patients with DWOP lesions was needed to detect a difference at the 0.05 level.

We analyzed a total of 50 from 138 subjects who had quantifiable DWOP lesions in SS-OCTA. The baseline data of the 50 patients for SS-OCTA quantitative analysis are given in Appendix A. As before, each eye was assessed in five different regions using the 12 mm × 12 mm scanning mode, and then we selected the images with DWOP lesions from these five 12 mm × 12 mm scanning images for the next analysis. Mirror-symmetric images of the healthy eye of the same subject were used as controls. This internal comparison between the two eyes of an individual allowed for greater control of most systemic confounding factors. Our SS-OCTA images analysis of the DWOP lesions had two main aspects:

We showed that the anatomical appearance of DWOP lesion areas on SS-OCTA en-face images at different layers, which corresponded to normal mirror-symmetric areas in the other eye of the same individual (Figure 4). We locally amplified and analyzed these areas of both eyes in SS-OCTA vascular en-face images of the inner retina layer, SVC, DVC, and choroid layer, and the SS-OCT thickness en-face images of all retina layers, the inner retina layer, the outer retina layer, and the choroid layer. Consequently, we were confident that there were no significant abnormalities except for the thickness en-face image of the outer retina layer, which showed darkening of the tissue visibly (Figure 4I1).

The other was that we selected the affected regions for quantitative analysis. The vessel density (VD) was defined as the ratio of the area of blood flow in the specific layer to the whole en-face scanning area in OCTA images. The three-dimensional (3D) choroidal vascularity index (CVI) was defined as the ratio of choroidal vascular luminal volume to total choroidal volume, reflecting the volumetric choroidal vascular density (Figure 3C,D) [19,20]. The mean thickness of the retina and choroid was calculated by dividing the total volume of the retina and choroid by 144 mm^2^. These parameters’ values were derived using built-in techniques developed by deep learning, referred to in previous research [19,21]. The raw image and color-coded map of each parameter was generated at each A-scan to illustrate them numerical in detail. The VD of the retinal, 3D CVI, and mean thickness of retinal and choroid values were also presented for grids of 1 mm × 1 mm (Figure 3E–G) [19]. Based on CFP and OCT en-face images, the border of the DWOP lesions was delineated in Photoshop CC 2018 software (Figure 5). Combined with OCTA B-scan images, we excluded the squares where the quality of the image was poor, then counted and calculated the average of the quantified values of all 1 mm × 1 mm squares where the DWOP lesion was covered in the affected eye. The values in the matching area of the contralateral eye were used as the control (Figure 5).

### 2.5. Reproducibility of SS-OCTA Measurements

Initially, the eyes of another 24 subjects were scanned twice by an experienced examiner (H.L.), and once again one week later by the same examiner. Then, the reproducibility of the SS-OCTA measurements was assessed using intraclass correlation coefficients (ICCs) and repeatability coefficients. Finally, the repeatability coefficient was computed as 1.96 times the standard deviation of the change between the two measurements, showing that the SS-OCTA measurements are highly reproducible. (Appendix A)

### 2.6. Statistical Analysis

Based on the SE, we divided the participants into three groups: SE ≥ −8, <−6 was group 1, SE ≥ −10, <−8 was group 2, and SE < −10 was group 3. A *t*-test (normal distribution) or Wilcoxon rank-sum test (non-normal distribution) were used for continuous variables; χ^2^ tests were used for categorical data (Table 1). Multivariate logistic regression models were used to determine if SE and AL correlated with DWOP or WWOP (Table 2). The Kruskal-Wallis rank-sum test was used to analyze the difference of VD in each layer and the CVI between eyes with lesions and contralateral healthy eyes (Table 3). All of the analyses were carried out in R (http://www.R-project.org) and the EmpowerStats software (www.empowerstats.com, X&Y solutions, Inc., Boston, MA, USA).

## 3. Results

We recruited 138 participants (227 eyes) in the cross-sectional study. The mean age was 24.18 ± 13.55 years old. The mean IOP was 15.99 ± 2.41 mmHg. The mean sphere, cylindrical, and SE were –7.65 ± 2.03, –1.25 ± 0.97, and –8.28 ± 2.21 D, respectively. The mean AL was 26.56 ± 1.10 mm. A total of 71.37% (162) of the participants were female.

### 3.1. Repeatability of Retinal and Choroidal Structural Measurement

The ICC values of topographical VD, CVI and Thickness varied from 0.905 to 0.999 for the SS-OCTA measurement repeatability in five different regions. The coefficients of repeatability varied from 0.11% to 1.02% for VD, from 1.07% to 2.81% for CVI, and from 6 to 19 μm for Thickness (Appendix A). Considering the means of these parameters, the repeatability of manual correction was good. The agreement between the twice SS-OCTA measurements, as assessed by ICCs and coefficients of repeatability, was also good.

### 3.2. The Incidence Rate of WWOP and DWOP

Table 1 demonstrates the characteristics of our cohort and a comparison of participants and different myopic groups. The incidence rate of WWOP and DWOP were 35.24% and 29.96%, respectively. Among the three myopic groups, SE (*p* < 0.001), AL (*p* < 0.001), and the prevalence of WWOP (*p* < 0.001) had significant differences. The patients with lower SE showed a tendency to have a higher AL and higher prevalence of WWOP. Age, IOP, gender, bilateral, and prevalence of DWOP showed no significant changes among the three myopic groups.

### 3.3. Risk Factors for WWOP and DWOP

Table 2 illustrates the associations of AL and SE with DWOP and WWOP using multivariate analysis. A greater AL (OR = 1.54, *p* = 0.0020) and a more serious SE (OR = 0.82, *p* = 0.0046) increased the risk for WWOP. A greater AL (OR = 2.09, *p* < 0.0001) and a more serious SE (OR = 0.84, *p* = 0.0153) also increased the risk for DWOP.

### 3.4. OCTA Quantitative Change in DWOP Lesions

Wilcoxon signed ranks tests were performed, as shown in Table 3, to compare the difference of OCTA quantitative indicators between eyes with DWOP and eyes without DWOP lesions. Since most of the lesions exceeded the OCTA scan range (including all WWOP lesions), 50 pairs of DWOP lesions were included in the analysis. The results showed that AL had no significant difference between eyes with DWOP and eyes without DWOP. Eyes with DWOP lesions had a lower VD in the whole retina (*p* < 0.001) and DVC layer (*p* < 0.001). The VD in the SVC layer and CVI values showed no significant difference between eyes with DWOP and eyes without DWOP. As for thickness, the thickness of the outer retina decreased in eyes with DWOP in comparison with healthy eyes (*p* < 0.001), whereas the thickness of the whole and inner retina and the choroid showed no significant change between the two groups.

## 4. Discussion

### 4.1. Incidence Rate and Risk Factors of WWOP and DWOP

Our study revealed that the higher the severity of myopia, the higher the incidence of WWOP, and AL and SE were independent protective factors for WWOP, which was consistent with the findings of previous studies [4,5,6]. Chen et al. [4] reported that the incidence of WWOP was 5.8% in emmetropia and 57.2% in people with high myopia. Zhang et al. [5] divided the myopia population into mild, moderate, and high myopia groups based on diopter –3.0 D and –6.0 D. The incidence of WWOP in those three groups was 10.9%, 21.5%, and 43.8%. The difference in the incidence rate among the three groups was statistically significant. Cheng et al. [6] reported that the incidence of WWOP among young people aged 12–18 with high myopia was 51.7%. Young people with peripheral retinopathy had a longer eye axis than those whose pathological changes were absent.

The existence of DWOP lesions has been reported in several diseases, but the incidence rate of DWOP lesions has rarely been reported [22,23,24]. In the case series reported by Fawzi A et al. [7] DWOP was found in a 25-year-old girl with myopia and showed no significant progress at the 1-year follow-up. Our data detailed the incidence rate of DWOP in highly myopic youths, and illustrated that both SE and AL are correlated with DWOP prevalence.

### 4.2. The Mechanisms of WWOP and DWOP

The mechanisms of WWOP and DWOP have been researched for decades, but remain a mystery. In the early stages of research, WWOP and DWOP were observed under ophthalmoscope and identified on account of a white or mottled brown retina without scleral indentation in fundus examination [14,15,25]. Considering that WWOP and DWOP always occur in the peripheral retina, and abnormal elongation of highly myopic eyes might create irregular curvatures and worsen the traction in the vitreoretinal interface, some researchers hypothesized that the reflectivity changes of these lesions may be associated with vitreoretinal stretching, which is similar to the mechanism for lattices and retinal holes [12,26].

With the development of OCT, peripheral lesions could be presented clearly in the vitreal, vitreal-retinal interface, retinal, and choroidal layers. In the study conducted by Amani A. Fawzi [7], WWOP was shown to have no relation to vitreal-retinal traction, but was related to the reflectivity changes in the photoreceptor layer, which is consistent with our research (shown in Figure 1). Previous studies have shown that DWOP can be seen in many fundus diseases, including hemoglobinopathy [15], retinal astrocytoma [7], white dot syndrome [7], HLA-B27 anterior middle uveitis [7], congenital retinal pigment epithelial hypertrophy [7,22], AIDS [19], Ebola [23], and choroidal osteoma [24]. The common feature of DWOP in OCT images is that the EZ is faded or has disappeared. Similarly, as a common lesion in myopic eyes, DWOP manifests as the lightening or disappearance of the photoreceptor cell layer in OCT.

### 4.3. New Insights Brought by Ultra-Wide SS-OCTA

With the advantages of deeper penetration, a wider field, and a faster scanning speed, SS-OCTA makes it possible to observe peripheral retinal lesions with better image quality and fewer artifacts. In addition, the high reproducibility of SS-OCTA measurements and the accuracy of peripheral retinal stratification guarantee the credibility of DWOP lesion quantification results. Based on quantitative data, we were able to analyze the change that happened in DWOP precisely. To our knowledge, this is the first study to apply ultra-wide SS-OCTA to analyze myopic peripheral changes. It is worth mentioning that we analyzed the perfusion of choroid middle and large vessels, a method which has emerged in recent years. As a new quantitative indicator, 3D CVI allows a good observation of the changes in choroid vascularity and new clinical insights [27,28], although the *p* value of 3D CVI in Table 3 was not less than 0.05. The DWOP lesion was characterized by thinning of the outer retina and decreased blood perfusion of the DVC layer in the lesion area. The potential correlation between photoreceptors and vascular changes has been described in previous studies of certain diseases [29,30,31]. Thus, we hypothesized that the explanation of our finding might be the degeneration of photoreceptors in the outer retina, leading to peripheral retinal atrophy and hypoxia. The vulnerable peripheral retina might be predisposed to further lesions, such as retinal breaks. However, further evidence for these speculations should be provided by long-term follow-up.

### 4.4. Limitations of the Current Study

There are several limitations to our study. Firstly, the cross-sectional nature of the study precludes any definite conclusions on the causality or temporal relationship, so further longitudinal studies are needed. Secondly, we performed a comparative analysis using DWOP lesion eyes versus healthy eyes. Although this internal comparison between the two eyes of an individual allows for greater control of most systemic confounding factors, there are still errors due to inconsistencies in the vascular pathways of the two eyes. Long-term follow-up is necessary for subsequent study and analysis in the same eye. Thirdly, we only focused on the young myopic population, and the generalizability of this result to other age groups is questionable. Finally, the main factor influencing measurements is image quality, which determines the accuracy of the peripheral retinal layering. Since WWOP is usually located at a more peripheral retinal location than DWOP and due to the limited scanning range of the currently used SS-OCTA instruments, it is difficult to obtain high-quality SS-OCTA images of the WWOP for subsequent quantitative analysis. However, for DWOP, the relatively high resolution and contrast of SS-OCTA images yielded good repeatability for retinal and choroidal segmentation.

## 5. Conclusions

In this study, we found that a more serious SE and a longer AL were risk factors for both WWOP and DWOP. We provided clear evidence of the interocular differences in DWOP lesion eyes and healthy eyes, showing that DWOP lesion eyes have lower perfusion in the DVC and a thinner outer retina for the first time. These findings offer some new insight into the role of the retina in myopic-related DWOP lesion development. Further longitudinal investigations would be valuable for testing the predictive value of decreased DVC blood flow and a thinner outer retina for DWOP lesions. Ultra-wide SS-OCTA can be considered a potential instrument in MRPD analysis due to its good reproducibility in observing the anatomical structure changes and quantifying vascularity variations.

## Figures and Tables

**Figure 1 jcm-11-04830-f001:**
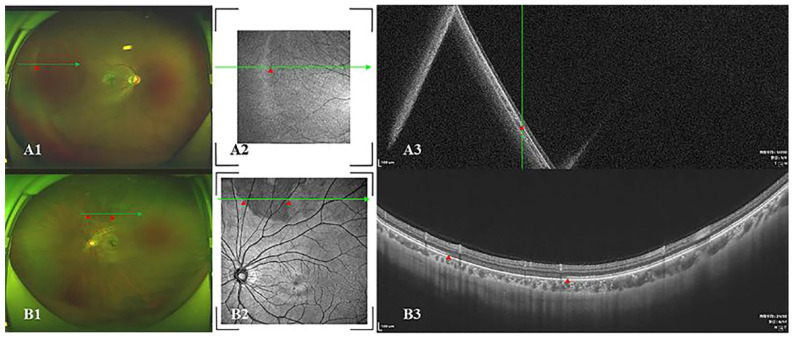
Multimode imaging of WWOP and DWOP. (**A1**–**A3**). One patient with WWOP. On the left of the red triangle, the ultra-wide color fundus photograph (**A1**) shows a hyperpigmented lesion region. The OCT en-face image was hyperreflective. The B-scan image shows that the ellipsoid zone is also hyperreflective. The WWOP is accompanied by DWOP on the right of the red triangle. (**B1**–**B3**). One patient with DWOP. Between the area of the two red triangles, the ultra-wide color fundus photograph (**B1**) shows a hypopigmented lesion region. The OCT en-face image is hyporeflective. The B-scan image shows that ellipsoid zone had faded or even disappeared. WWOP, white without pressure; DWOP, dark without pressure; OCT, optical coherence tomography.

**Figure 2 jcm-11-04830-f002:**
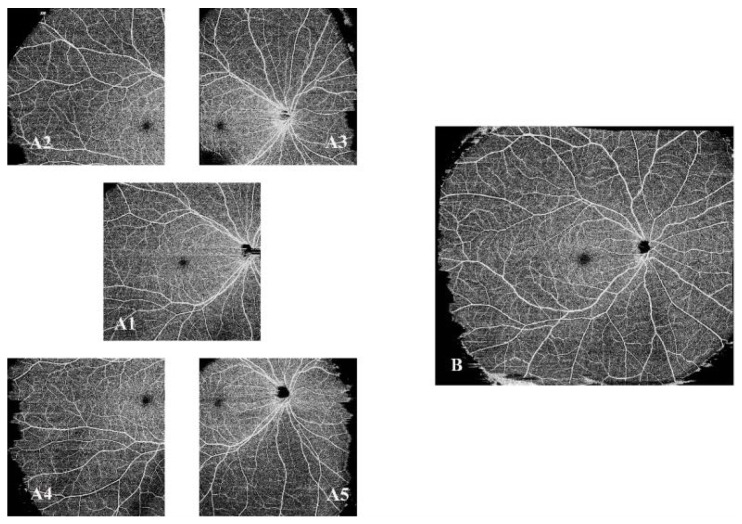
Swept source optical coherence tomography angiography (SS-OCTA) images of the right eye. (**A1**–**A5**) SS-OCTA images of 12 mm × 12 mm scanning mode in different regions. (**A1**) Macula-centered. (**A2**) Superior temporal. (**A3**) Superior nasal. (**A4**) Inferior temporal. (**A5**) Inferior nasal. (**B**). The composite image of (**A1**–**A5**) with an area of 23.5 mm × 17.5 mm. SS-OCTA, swept-source optical coherence tomography angiography.

**Figure 3 jcm-11-04830-f003:**
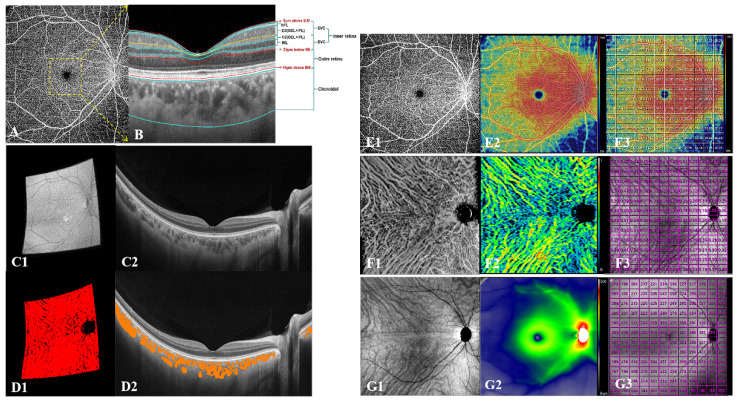
SS-OCTA imaging and quantification. (**A**) 12 mm × 12 mm SS-OCTA scanning. (**B**) 3 mm × 3 mm OCT image of the yellow square in (**A**). Blue, red, and yellow curves represent segmentation layers. ILM, internal limiting membrane; NFL, nerve fiber layer; GCL, ganglion cell layer; IPL, inner plexiform layer; INL, inner nuclear layer; BM, Bruch’s membrane; SVC, superficial vascular complex; DVC, deep vascular complex. (**C1**–**D2**) 3D CVI pictures derived using built-in techniques. The orange areas in (**D2**) represent choroidal vascular luminal areas in one B-scan layer. (**D1**) shows the synthesized choroidal vascular luminal volume, which was derived from 1024 B-scans (such as (**D2**)). (**E1**–**G1**) The raw images of VD, CVI, and retinal thickness. (**E****2**–**G****2**) The color-coded maps of VD, CVI, and retinal thickness. (**E****3**–**G****3**) The mean VD, CVI, and retinal thickness presented in 1 × 1 mm grids. SS-OCTA, swept-source optical coherence tomography angiography; OCT, optical coherence tomography; 3D CVI, three-dimensional choroidal vascularity index; VD, vessel density.

**Figure 4 jcm-11-04830-f004:**
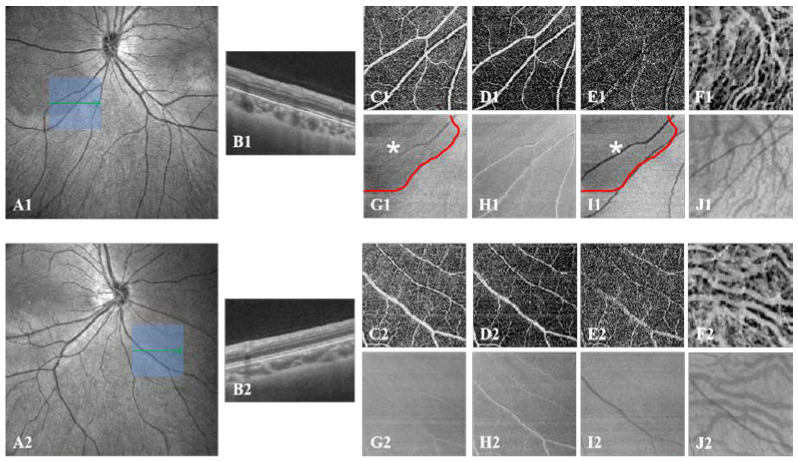
SS-OCT/OCTA images of local lesion areas and corresponding normal areas in both eyes. (**A1**–**J1**) Images of one eye with a DWOP lesion. (**A1**) SS-OCT en-face image of the left eye, revealing the location (blue cube) from which all subsequent images (**C1**–**J1**) were taken and the single scan (green arrow) which corresponds to the OCT B-scan (**B1**); (**C1**–**F1**) SS-OCTA vascular en-face images of the inner retina, SVC, DVC, and choroid; (**G1**–**J1**) SS-OCT thickness en-face images of all retina layers, inner retina, outer retina, and choroid. DWOP lesion (shown as a red curve) in (**G1**) and (**I1**) (white *). (**A2**–**J2**) Images of the comparison eye without a DWOP lesion. (**A2**) SS-OCT en-face image of the right eye, revealing the location (blue cube) from which all subsequent images (**C2**–**J2**) were taken, and the single scan (green arrow), which corresponds to the OCT B-scan (**B2**); (**C2**–**F2**) SS-OCTA vascular en-face images of the inner retina, SVC, DVC, and choroid; (**G2**–**J2**) SS-OCT thickness en-face images of all retina layers, inner retina, outer retina, and choroid, which appear to be normal. DWOP, dark without pressure; SS-OCTA, swept-source optical coherence tomography angiography; OCT, optical coherence tomography; SVC, superficial vascular complex; DVC, deep vascular complex.

**Figure 5 jcm-11-04830-f005:**
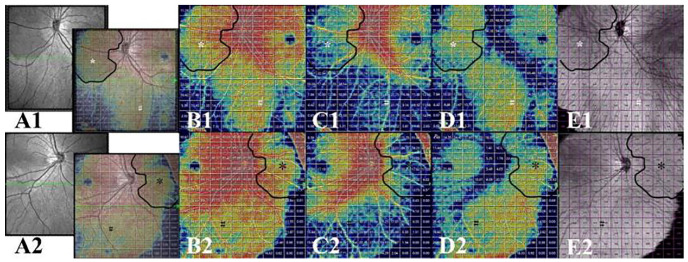
Quantification of all lesion areas and corresponding normal areas in both eyes. (**A1**–**E1**) Images of one eye with a DWOP lesion. The DWOP lesion was manually delineated based on en-face OCT images (shown as a black curve). VD (**B1**–**D1**) and CVI values (**E1**) presented in 1 mm × 1 mm grids were used in the statistical analysis. The white * represents the site of the DWOP lesion, the white # represents the normal area. (**A2**–**E2**) Images of the compared eye without a DWOP lesion. The black curve was obtained by horizontally flipping the black curve in (**A1**–**E1**) to the corresponding position. VD (**B2**–**D2**) and CVI values (**E2**) presented in 1 mm × 1 mm grids were used in the statistical analysis. The black * was used for matching analysis with the DWOP lesion (white *) in (**A1**–**E1**). The black # was used for matching analysis with the normal area (white #) in (**A1**–**E1**). DWOP, dark without pressure; OCT, optical coherence tomography; VD, vessel density; CVI, choroidal vascularity index.

**Table 1 jcm-11-04830-t001:** Characteristics and comparison of participants in different groups.

Variables	Overall	Group 1	Group 2	Group 3	*p* Value
		SE ≥ −8, <−6	SE ≥ −10, <−8	SE < −10	
No. of eyes	227	130	62	35	
Age, years	24.18 ± 13.55	24.62 ± 17.82	23.44 ± 2.43	23.89 ± 1.78	0.067
IOP, mmHg	15.99 ± 2.41	15.94 ± 2.38	15.98 ± 2.18	16.18 ± 2.96	0.825
SE, D	−8.28 ± 2.21	−6.93 ± 0.58	−8.82 ± 0.50	−12.29 ± 2.61	<0.001 ^a^
Sphere, D	−7.65 ± 2.03	−6.47 ± 0.69	−8.10 ± 0.60	−11.26 ± 2.40	<0.001 ^a^
Cylindrical, D	−1.25 ± 0.97	−0.94 ± 0.70	−1.45 ± 0.97	−2.06 ± 1.24	<0.001 ^a^
AL, mm	26.56 ± 1.10	26.12 ± 0.85	26.73 ± 0.66	27.93 ± 1.36	<0.001 ^a^
Gender					0.967
Female	162 (71.37%)	92 (70.77%)	45 (72.58%)	25 (71.43%)	
Male	65 (28.63%)	38 (29.23%)	17 (27.42%)	10 (28.57%)	
Bilateral					0.739
OD	116 (51.10%)	65 (50.00%)	31 (50.00%)	20 (57.14%)	
OS	111 (48.90%)	65 (50.00%)	31 (50.00%)	15 (42.86%)	
WWOP					<0.001 ^a^
No	147 (64.76%)	91 (70.00%)	45 (72.58%)	11 (31.43%)	
Yes	80 (35.24%)	39 (30.00%)	17 (27.42%)	24 (68.57%)	
DWOP					0.121
No	159 (70.04%)	97 (74.62%)	42 (67.74%)	20 (57.14%)	
Yes	68 (29.96%)	33 (25.38%)	20 (32.26%)	15 (42.86%)	

*p* values for differences in variables among group 1, group 2, and group 3 were based on the Wilcoxon rank-sum test or chi-square test, as appropriate. ^a^ statistical significance. SE, spherical equivalent; IOP, intraocular pressure assessment; AL, axial length; OD, oculus dexter; OS, oculus sinister; WWOP, white without pressure; DWOP, dark without pressure.

**Table 2 jcm-11-04830-t002:** Associations of AL and SE with DWOP and WWOP.

Exposure	Non-Adjusted		Adjusted	
	OR (95%CI)	*p* Value	OR (95%CI)	*p* Value
WWOP				
AL	1.33 (1.03, 1.71)	0.0263 ^a^	1.54 (1.17, 2.03)	0.0020 ^a^
SE	0.83 (0.73, 0.95)	0.0075 ^a^	0.82 (0.72, 0.94)	0.0046 ^a^
DWOP				
AL	2.03 (1.49, 2.76)	<0.0001 ^a^	2.09 (1.51, 2.91)	<0.0001 ^a^
SE	0.84 (0.74, 0.96)	0.0129 ^a^	0.84 (0.73, 0.97)	0.0153 ^a^

ORs were derived from multivariate logistic regression analysis. We adjusted the model for age, gender, bilateral, and IOP. ^a^ statistical significance. CI, confidence interval. WWOP, white without pressure; AL, axial length; SE, spherical equivalent; DWOP, dark without pressure.

**Table 3 jcm-11-04830-t003:** Quantitative analysis of different regions in paired eyes using SS-OCTA.

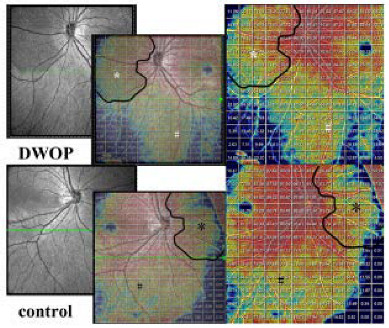	**Variables** **(Control-DWOP)**	*** Area**		**# Area**	
		***Z* Value**	***p* Value**	***Z* Value**	***p* Value**
	AL	−0.556	0.535	−0.556	0.535
	VD				
	Inner retina	5.873	<0.001 ^a^	1.236	0.106
	SVC	−1.132	0.159	−0.853	0.302
	DVC	5.231	<0.001 ^a^	1.568	0.091
	CVI	−0.463	0.628	0.168	0.970
	Thickness				
	All retina	1.098	0.175	−0.737	0.413
	Inner retina	−0.791	0.384	−0.363	0.729
	Outer retina	6.114	<0.001 ^a^	1.676	0.086
	Choroid	−0.432	0.662	−0.998	0.201

Z values and *p* values were derived from the Kruskal-Wallis rank sum test. ^a^ statistical significance. As illustrated in Figure 5, the * in Table 3 represents the site of the DWOP lesion area and its control area. And the # in Table 3 represents the normal area and its control area. DWOP, dark without pressure; AL, axial length; VD, vessel density; SVC, superficial vascular complex; DVC, deep vascular complex; CVI, choroidal vascularity index.

## Data Availability

The data presented in this study are available on request from the corresponding author. The data are not publicly available due to privacy and ethical reasons.

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
