# Peer review of "Analysis of White and Dark without Pressure in a Young Myopic Group Based on Ultra-Wide Swept-Source Optical Coherence Tomography Angiography"

_jcm, 2022, doi:10.3390/jcm11164830_

Round 1

Reviewer 1 Report

The authors evaluated the areas of WWOP and DWOP in myopic eyes. Some part of the investigation is related to incidence and association with axial length and myopia. the more interesting part is related to wide-field OCT and OCTA data. It is unclear to me if the imaging of these areas were compared between fellow eyes.

The terminology of "hypo" and "hyper" reflective is usually used for infrared images rather than fundus photos.probably, "hypo' and "hyper" pigmented lesions are more appropriate.

Various vascular layers of retina and choroid may be hard to define in periphery; so OCTA quantitative data may not be that much accurate in that far periphery areas of retina, subjective to measurement errors. Comparing this between an eye with lesion with a healthy eye may not be a good way. How did the authors match for this comparison? This should be mentioned in the limitation part.

There is no wrap up paragraph at the end of the manuscript, making hard to draw the conclusion about the impact or clinical implication of this study.

There are a few sentences need rewording:

line 210-212: " In the study conducted by Amani A. Fawzi 5, WWOP was shown to have no relation with vitreal-retinal traction, but the reflectivity changes in the photoreceptor layer, which is consistent with our research"

line 214: "white spot syndrome" should be " white dot syndrome"?

Author Response

Reply to Reviewer 1

We would like to appreciate your careful reading, helpful comments, and constructive suggestions, which has significantly improved the presentation of our manuscript.

We have carefully considered all the comments from you and revised our manuscript accordingly. We have combed our paper and rewritten the Introduction, Method, Result and Discussion parts. And the reference list of the revised manuscript covered the relevant literature adequately. Follows are our responses to your questions. We really hope that our responses have well addressed all the concerns from you and our revised manuscript can be accepted for publication.

  1. The authors evaluated the areas of WWOP and DWOP in myopic eyes. Some part of the investigation is related to incidence and association with axial length and myopia. the more interesting part is related to wide-field OCT and OCTA data. It is unclear to me if the imaging of these areas were compared between fellow eyes.

Thank you for the above comment. The imaging of these DWOP areas was compared with the patient's own mirror-symmetrical areas of the other non-lesioned eye. (Page 4 Line 145-147 )

  1. The terminology of "hypo" and "hyper" reflective is usually used for infrared images rather than fundus photos.probably, "hypo' and "hyper" pigmented lesions are more appropriate.

Thank you for your suggestion and we fully endorse that. We have changed into    " ‘hypo’ and ‘hyper’ pigmented lesions" in our revised manuscript. (Page 2 Line 87, 89 ; Figure1)

  1.  
  • Various vascular layers of retina and choroid may be hard to define in periphery; so OCTA quantitative data may not be that much accurate in that far periphery areas of retina, subjective to measurement errors.
  • Comparing this between an eye with lesion with a healthy eye may not be a good way. How did the authors match for this comparison? This should be mentioned in the limitation part.

  • Thank you for your rigorous consideration. However, we are relatively confident about the accuracy of the stratification and quantification of the peripheral region using this SS-OCTA device, and we have presented an image with stratification details of the peripheral region in the supplemental file. ( Supplementary Figure 1 )

  • We totally understand your concern. Indeed, we compared this between an eye

with DWOP lesion with a healthy eye, which may have some limitations, but this is the best way we have come up with so far.

There are two points that support this:

  1. We compared the DWOP lesion eye with the other healthy eye from the same subject. This internal comparison between the two eyes of an individual allows for greater control of most systemic confounding factors.
  2. The health area we selected for comparison is mirror symmetrical to the DWOP lesion area, which minimizes the error due to inconsistent vascular pathways.

The mirror-symmetric areas were acquired in two steps: 

  1. First, we acquired 12mm x 12mm SS-OCTA images of the corresponding areas of the left and right eyes from the mirror-symmetric internal fixation eye positions. 
  2. Then, we performed a mirror flip on the PS software and outlined the areas that were to be quantified in the next step.

  1. There is no wrap up paragraph at the end of the manuscript, making hard to draw the conclusion about the impact or clinical implication of this study.

Thank you for your professional advice. We have added the Conclusion part in our revised manuscript as follows:

Conclusion:

In this study, we found that a more serious SE and a longer AL were risk factors for both WWOP and DWOP. We provided clear evidence of the interocular differences in DWOP lesion eyes and healthy eyes, showing that DWOP lesion eyes have lower perfusion in the DVC and a thinner outer retina for the first time. These findings offer some new insight into the role of the retina in myopic-related DWOP lesion development. Further longitudinal investigations would be valuable for testing the predictive value of decreased DVC blood flow and a thinner outer retina for DWOP lesions. Ultra-wide SS-OCTA can be considered a potential instrument in MRPD analysis because of its good reproducibility in observing the anatomical structure changes and quantifying the vascularity variations. (Page 11 Line 367-377 )

  1. There are a few sentences need rewording:
  • line 210-212: " In the study conducted by Amani A. Fawzi 5, WWOP was shown to have no relation with vitreal-retinal traction, but the reflectivity changes in the photoreceptor layer, which is consistent with our research" 
  • line 214: "white spot syndrome" should be " white dot syndrome"?

We apologize for the incorrect writing.

1) We have reworded this sentence into “ In the study conducted by Amani A. Fawzi 7, WWOP was shown to have no relation to vitreal-retinal traction, but was related to the reflectivity changes in the photoreceptor layer, which is consistent with our research (shown in Figure 1).“ (Page 10 Line 322-325 )

2) We have changed  "white spot syndrome" into " white dot syndrome" in our revised manuscript. (Page 10 Line 327 )

Reviewer 2 Report

The authors presented an interesting study investigating  the relationship between white without pressure and dark without pressure in myopic subjects and the quantitative changes assessed by ultra-wide swept-source optical coherence tomography angiography. The content of the manuscript is with merit and the findings could be worth reporting but the quality of presentation is low and the English language (in term of grammar, stile, use of punctuation, vocabulary) is very poor. The entire manuscript should be revised and improved before publication can be considered. In addition, the authors should address the following comments before publication.

Abstract

-       Line 13-14: please rephrase in order to clarify what is the purpose of the study (if there are two separate aims please indicate so)

Entire manuscript

-       The English language (in term of grammar, stile, use of punctuation) is very poor. The entire manuscript should be revised and improved before publication can be considered. 

Methods

-       Line 64: what the author mean by “normal intraocular pressure”? 

-       Statistics: The author should provide a statistical power estimation for their study or at least some justification of the study n and add it to the methods 

-       The authors state for the first time in the methods that they did also a reproductivity evaluation of the SS-OCTA measurements – this information should be provided in the introduction first and then presented also in the results and discussion sections.

-       The specific details about number of patients/images analyzed in the study and type of analysis should be clearly stated in the methods section.

Discussion

-       The authors should add a “limitation” section

-       The authors should add a paragraph “conclusion” at the end of the manuscript clearly stating the conclusions derived from the results of their study, the potential clinical applications of their results in term of everyday practice, and highlight the future direction of research in this field

Author Response

Reply to Reviewer 2

We appreciate the time and effort that you dedicated to providing feedback on our manuscript, and we are grateful for the insightful comments and valuable improvements to our manuscript. After studying these comments carefully, we have combed our paper and rewritten the Introduction, Method, Result, and Discussion parts, which we hope will meet with approval. And the reference list of the revised manuscript covered the relevant literature adequately. The revised manuscript has also been double-checked, and the typos and grammar errors have been corrected by a native English speaker (Certification Attached). The main corrections in the manuscript and the responses to your comments are as follows:

  1. The authors presented an interesting study investigating  the relationship between white without pressure and dark without pressure in myopic subjects and the quantitative changes assessed by ultra-wide swept-source optical coherence tomography angiography. The content of the manuscript is with merit and the findings could be worth reporting but the quality of presentation is low and the English language (in term of grammar, stile, use of punctuation, vocabulary) is very poor. The entire manuscript should be revised and improved before publication can be considered. In addition, the authors should address the following comments before publication.

Thank you for your comments. The revised manuscript has also been double-checked, and the typos and grammar errors have been corrected by a native English speaker (Certification Attached).

  1. Abstract

Line 13-14: please rephrase in order to clarify what is the purpose of the study (if there are two separate aims please indicate so)

We apologize for the insufficiently rigorous writing. We have rewritten it as “To investigate the incidence of white without pressure (WWOP) and dark without pressure (DWOP) in a young myopic group based on multimode imaging. And to explore the quantitative changes in DWOP based on ultra-wide swept-source optical coherence tomography angiography (SS-OCTA).” in our revised manuscript. (Page 1 Line 14-17 )

  1. Entire manuscript

The English language (in term of grammar, stile, use of punctuation) is very poor. The entire manuscript should be revised and improved before publication can be considered. 

We apologize for the poor writing, The revised manuscript has been double-checked, and the typos and grammar errors have been corrected by a native English speaker  (Certification Attached).

  1. Methods

1) Line 64: what the author mean by “normal intraocular pressure”? 

Thank you for your comments. "Normal intraocular pressure"means that the intraocular pressure (IOP) is 12-21±2 mmHg, and the difference in IOP between the two eyes of the same subject is ≤5mmHg. (Page 2 Line 73-74 )

Reference:

[1] Allingham RR, Damji K, Freedman S, et al. Shields' Textbook of Glaucoma, in Pine J and Murphy J (eds). Philadelphia, Lippincott Williams & Wilkins, 2005, ed 5, pp 197-207.

2) Statistics: The author should provide a statistical power estimation for their study or at least some justification of the study n and add it to the methods 

We are grateful for your proposal. We have added this content to our manuscript as follows:

The DWOP lesion sample size was determined by np=[(Zα+Zβ)  σd/ ES]2 18 in the Power Analysis and Sample Size (PASS) software 2020, using preliminary data obtained in our laboratory with the following assumptions: an α (type I error p value significance level) of 0.05 (two-tailed), a power (1-β) of 90%, difference (ES) in the thickness of all retina between the DWOP lesion area and healthy control area of 6 μm, and a standard deviation (σd) of 10 μm. Considering the situation of loss of follow-up, we set the loss ratio of follow-up to 20%. Therefore, a minimum of 37 patients with DWOP lesions was needed to detect a difference at the 0.05 level. (Page 4 Line 132-140 )

Reference:

[18] Malone HE, Nicholl H, Coyne I. Fundamentals of estimating sample size. Nurse Res. 2016;23(5):21-25.

3) The authors state for the first time in the methods that they did also a reproductivity evaluation of the SS-OCTA measurements -- this information should be provided in the introduction first and then presented also in the results and discussion sections.

Thank you, and we do agree with your suggestions and have improved this content in the Introduction, Results, and Discussion sections. And detailed results are presented in Supplementary Table 2. 

Introduction

  Its high-quality imaging and reproducible scanning make it possible to quantify the retina and choroid 13.  (Page 2 Line 60-61 )

Results

Repeatability of Retinal and Choroidal Structural Measurement

The ICC values of topographical VD, CVI and Thickness varied from 0.905 to 0.999 for the SS-OCTA measurement repeatability in five different regions. The coefficients of repeatability varied from 0.11% to 1.02% for VD, from 1.07% to 2.81% for CVI, and from 6 to 19 μm for Thickness (Supplemenary Table 2). Considering the means of these parameters, the repeatability of manual correction was good. The agreement between the twice SS-OCTA measurements, as assessed by ICCs and coefficients of repeatability, was also good.  (Page 8 Line 258-266 )

Discussion

In addition, the high reproducibility of SS-OCTA measurements and the accuracy of peripheral retinal stratification guarantee the credibility of DWOP lesion quantification results. (Page 10 Line 335-337 )

Reference:

[13] Wu H, Xie Z, Wang P, et al. Differences in Retinal and Choroidal Vasculature and Perfusion Related to Axial Length in Pediatric Anisomyopes. Invest Ophthalmol Vis Sci. 2021;62(9):40.

4) The specific details about number of patients/images analyzed in the study and type of analysis should be clearly stated in the methods section.

Thank you for your reminder, we have sorted out this part in detail and made a comprehensive revision, the specific details and the method of analysis have been marked clearly. And the baseline data of the 50 patients for SS-OCTA quantitative analysis is in Supplementary Table 1.

We analyzed a total of 50 from 138 subjects who had quantifiable DWOP lesions in SS-OCTA. The baseline data of the 50 patients for SS-OCTA quantitative analysis are given in Supplementary Table 1. As before, each eye was assessed in five different regions using the 12 mm×12 mm scanning mode, and then we selected the images with DWOP lesions from these five 12 mm×12 mm scanning images for the next analysis. Mirror-symmetric images of the healthy eye of the same subject were used as controls. This internal comparison between the two eyes of an individual allowed for greater control of most systemic confounding factors. Our SS-OCTA images analysis of the DWOP lesions had two main aspects: (Page 4 Line 141-149 )

  1. Discussion

1) The authors should add a “limitation” section

- 2) The authors should add a paragraph “conclusion” at the end of the manuscript clearly stating the conclusions derived from the results of their study, the potential clinical applications of their results in term of everyday practice, and highlight the future direction of research in this field

Thank you for your professional advice. We have added the Limitation section and Conclusion part in our revised manuscript as follows:

  • Limitations of the Current Study

There are several limitations to our study. Firstly, the cross-sectional nature of the study precludes any definite conclusions on the causality or temporal relationship, so further longitudinal studies are needed. Secondly, we performed a comparative analysis using DWOP lesion eyes versus healthy eyes. Although this internal comparison between the two eyes of an individual allows for greater control of most systemic confounding factors, there are still errors due to inconsistencies in the vascular pathways of the two eyes. Long-term follow-up is necessary for subsequent study and analysis in the same eye. Thirdly, we only focused on the young myopic population, and the generalizability of this result to other age groups is questionable. Finally, the main factor influencing measurements is image quality, which determines the accuracy of the peripheral retinal layering. However, the relatively high resolution and contrast of our images yielded good repeatability for retinal and choroidal segmentation. (Page 10 Line 353-365 )

  • Conclusion

In this study, we found that a more serious SE and a longer AL were risk factors for both WWOP and DWOP. We provided clear evidence of the interocular differences in DWOP lesion eyes and healthy eyes, showing that DWOP lesion eyes have lower perfusion in the DVC and a thinner outer retina for the first time. These findings offer some new insight into the role of the retina in myopic-related DWOP lesion development. Further longitudinal investigations would be valuable for testing the predictive value of decreased DVC blood flow and a thinner outer retina for DWOP lesions. Ultra-wide SS-OCTA can be considered a potential instrument in MRPD analysis because of its good reproducibility in observing the anatomical structure changes and quantifying the vascularity variations. (Page 11 Line 367-377 )

Round 2

Reviewer 1 Report

Intro: there are no strong evidence that lattice requires prophylaxis barrier laser

conclusion and title: the authors wanted to talk about both DWOP and WWOP, but most of their focus of findings is on DWOP

Discussion: "As a new quantitative indicator, 3D CVI allows a good observa tion of the changes in choroid vascularity and new clinical insights, although the result we obtained was negative." it is not clear what part of their result was negative

Author Response

Reply to Reviewer 1

We appreciate the time and effort that you dedicated to providing feedback on our manuscript, and we are grateful for the insightful comments and valuable improvements to our manuscript. The revised manuscript has also been double-checked, and the typos and grammar errors have been corrected by a native English speaker (Certification Attached). Revisions made to the manuscript have been marked up using the "Track Changes" function and distinguished in red font.

The main corrections in the manuscript and the responses to your comments are as follows:

  1. Intro: there are no strong evidence that lattice requires prophylaxis barrier laser

 Thank you for your rigorous comments. Indeed, according to Wilkinson [1], there is no strong evidence to demonstrate the effectiveness of prophylactic therapy in eyes with asymptomatic retinal breaks or lattice degeneration, for which the evidence is current to February 2014. But to our knowledge, subsequent studies have also shown that early prophylactic laser treatment warrants sincere consideration in myopic eyes of peripheral lattice degeneration.[2] However, for a more rigorous presentation, we have made the following changes to the original manuscript:

“For example, lattice degeneration is relatively common and may require prophylactic laser retinopexy, which is associated with retinal atrophy and may lead to retinal detachment 3.(Page 1 Line 40-42)

Reference:

  • Wilkinson CP. Interventions for asymptomatic retinal breaks and lattice degeneration for preventing retinal detachment. Cochrane Database Syst Rev. 2014;2014(9):CD003170. Published 2014 Sep 5.
  • Tsai CY, Hung KC, Wang SW, Chen MS, Ho TC. Spectral-domain optical coherence tomography of peripheral lattice degeneration of myopic eyes before and after laser photocoagulation. J Formos Med Assoc. 2019;118(3):679-685.

  1. conclusion and title: the authors wanted to talk about both DWOP and WWOP, but most of their focus of findings is on DWOP

 Thank you for your comments. Since WWOP is usually located at a more peripheral retinal location than DWOP and due to the limited scanning range of the currently used SS-OCTA instruments, it is difficult to obtain high-quality SS-OCTA images of the WWOP for subsequent quantitative analysis. We have added this to the limitation part of our manuscript, based on helpful comments from the reviewer.

Finally, the main factor influencing measurements is image quality, which determines the accuracy of the peripheral retinal layering. Since WWOP is usually located at a more peripheral retinal location than DWOP and due to the limited scanning range of the currently used SS-OCTA instruments, it is difficult to obtain high-quality SS-OCTA images of the WWOP for subsequent quantitative analysis. However, for DWOP, the relatively high resolution and contrast of SS-OCTA images yielded good repeatability for retinal and choroidal segmentation.(Page 10 Line 365-368)

  1. Discussion: "As a new quantitative indicator, 3D CVI allows a good observation of the changes in choroid vascularity and new clinical insights, although the result we obtained was negative." it is not clear what part of their result was negative

 Thank you very much for your precious comments. The meaning of “the result we obtained was negativeis that the P value of 3D CVI in the Kruskal-Wallis rank sum test of Table 3 was not less than 0.05, indicating no significant statistical significance. To avoid confusing understanding, we have revised this sentence in the manuscript as follows:

"As a new quantitative indicator, 3D CVI allows a good observation of the changes in choroid vascularity and new clinical insights, although the P value of 3D CVI in Table 3 was not less than 0.05."(Page 10 Line 342-344)

Reviewer 2 Report

The authors addressed most of my comments, but some additional comments need to be addressed:

- In the abstract, please revise as follows lines 14-15: “To investigate the incidence of white without pressure (WWOP) and dark without pressure (DWOP) in a young myopic group based on multimode imaging; and to explore the quantitative changes in DWOP based on ultra-wide swept-source optical coherence tomography angiography (SS-OCTA).” 

- at the end of the introduction, the authors should clearly state that they also performed an evaluation of the reproducibility of the SS-OCTA measurements. As the manuscript currently stands, this information is provided in the methods (lines 22-228) but the authors should indicate already in the introduction that they also aimed to evaluate the reproducibility of the measurements.

- the authors evaluate the intra-operator reproducibility of the technique - was the inter-operator reproducibility evaluated? 

- there are still some style issues to correct: for example why lines 280-281 are in bold characters?

Author Response

Reply to Reviewer 2

We would like to appreciate your careful reading, helpful comments, and constructive suggestions, which has significantly improved the presentation of our manuscript. And we have carefully considered all the comments from you and revised our manuscript accordingly. Revisions made to the manuscript have been marked up using the "Track Changes" function and distinguished in red font.

Follows are our responses to your questions. We really hope that our responses have well addressed all the concerns from you and our revised manuscript can be accepted for publication.

  1. - In the abstract, please revise as follows lines 14-15: “To investigate the incidence of white without pressure (WWOP) and dark without pressure (DWOP) in a young myopic group based on multimode imaging; and to explore the quantitative changes in DWOP based on ultra-wide swept-source optical coherence tomography angiography (SS-OCTA).” 

Thank you for your professional advice. We have revised it as proposed above. (Page 1 Line 14-17)

  1. - at the end of the introduction, the authors should clearly state that they also performed an evaluation of the reproducibility of the SS-OCTA measurements. As the manuscript currently stands, this information is provided in the methods (lines 22-228) but the authors should indicate already in the introduction that they also aimed to evaluate the reproducibility of the measurements.

We really appreciate your suggestions and have added this part to the Introduction. As follows:

Thus, this study aimed to investigate the incidence of WWOP and DWOP in young adults, evaluate the reproducibility of the SS-OCTA measurements, and explore the quantitative changes of lesions based on ultra-wide SS-OCTA.(Page 2 Line 61-64)

  1. - the authors evaluate the intra-operator reproducibility of the technique - was the inter-operator reproducibility evaluated? 

Thank you for your professional comments. Before formally recruiting patients, we did also evaluate the inter-operator reproducibility. Good agreement between the examiners is a prerequisite for us to start subsequent experiments. Regrettably, owing to space constraints, we didn't show and illustrate this part of the data in our manuscript. The detailed operation steps are as follows:

First, 48 eyes from another 24 subjects were selected, and these 12mm×12mm SS-OCTA images were scanned and segmented twice by two trained examiners (H.L. and H.M.Y.). Then, intraclass correlation coefficients (ICCs) and coefficients of repeatability were calculated to assess the agreement of VD, CVI and Thickness measurements within and between examiners at five various regions. The coefficient of repeatability was calculated as 1.98 times the standard deviation of the differences between two measurements. After obtaining good agreement, all the scans were measured by the same examiner (H.L.).

  1. - there are still some style issues to correct: for example why lines 280-281 are in bold characters?

We apologize for the insufficiently rigorous writing format. We have carefully rechecked the entire manuscript and revised the insufficiently rigorous writing. (Page 9 Line 284-285)